# Exposure of the African mound building termite, *Macrotermes bellicosus* workers to commercially formulated 2,4-D and atrazine caused high mortality and impaired locomotor response

**Afure J. Ejomah**[1☯], **Osariyekemwen O. Uyi**[1,2☯¤*], **Sese-Owei Ekaye**[1☯]

**1** Department of Animal and Environmental Biology, University of Benin, Benin City, Nigeria, **2** Department of Zoology and Entomology, University of Fort Hare, Alice, South Africa

☯ These authors contributed equally to this work.
¤ Current address: Department of Ecosystem Science and Management, Pennsylvania State University, University Park, Pennsylvania, United States of America
* osariyekemwen.uyi@uniben.edu

**Data Availability Statement:** All relevant data are within the manuscript.

## Abstract

Recent empirical evidence suggests that herbicides have damaging effects on non-target organisms in both natural and semi-natural ecosystems. The African mound building termite, *Macrotermes bellicosus*, is an important beneficial insect that functions as an ecosystem engineer due to its role in the breakdown of dead and decaying materials. Here, we examined the effects of 2,4-D amine salt (2,4-D) and atrazine based herbicides viz. Vestamine® and Ultrazine® on the survival and locomotion response of *M. bellicosus*. Worker termites were treated with a range of concentrations of Vestamine® (the recommended concentration: 6.25 ml per 500 ml of water, 0.25- and 0.5-fold below the recommended concentration and distilled water as control) and Ultrazine® (the recommended concentration: 3.75 ml per 500 ml of water, 0.25-, 0.5-, 2.0- and 4-fold of the recommended concentration and distilled water as control) for 24 hours for the mortality test, and allowed to run for 15 seconds for the locomotion trial. All concentrations of both Vestamine® and Ultrazine® were highly toxic to worker termites and mortality increased as the concentration and time after treatment increased. For both herbicides, concentrations far less than the recommended rates caused 100% mortality. The speed of termites was significantly influenced by both Vestamine® and Ultrazine® as termites exposed to all tested concentrations of the herbicides exhibited reduced running speed than the control. These findings suggest that beneficial insects, especially *M. bellicosus* may experience high mortality (up to 100%) and reduced mobility if they are sprayed upon or come in contact with plant materials that have been freshly sprayed with (less or more than) the recommended concentrations of Vestamine® and Ultrazine®. The findings of our study calls for the reassessment of the usage of 2,4-D and atrazine based herbicides in weed control in termite and other beneficial insect populated habitats.

**Funding:** The author(s) received no specific funding for this work.

**Competing interests:** The authors have declared that no competing interests exist.

## Introduction

Pesticides are ubiquitously considered the major driver of the loss of important beneficial insects. Only 10% of pesticides applied for pest control reaches the target organisms while the remaining bulk contaminates the environment where they adversely affect public health and beneficial organisms [1,2]. Globally, herbicides are the most widely used form of pesticides [3]. In tropical regions including Nigeria, intensive agricultural practices in order to increase yield has led to an increase in the use of agrochemicals such as herbicides [4]. Of a range of herbicides present in Nigeria, 2,4-Dichlorophenoxyacetic acid (2,4-D) and atrazine based herbicides rank among the most widely used selective weed control chemicals where locals in both rural and peri-urban communities rampantly use these chemical purposely for controlling weeds in farms and around their homes [5, O. Uyi, Personal observation.]. The locals opt for 2,4-D and atrazine based herbicides because these herbicides (i) kill most annual and perennial pre-emergent, emergent and post-emergent weeds and (ii) are cheaper and less laborious to use than manual or mechanical weed control methods (O. Uyi, Personal observation.).

2,4-D is a systemic herbicide and the first successful selective broadleaf plant herbicide, allowed for weed control in wheat (*Triticum aestivum*), corn (*Zea mays*), and rice (*Oryza sativa*) [6]. 2,4-D is a selective systemic herbicide that acts by mimicking the action of the plant growth hormone auxin, by stimulating growth, rejuvenating old cells and over stimulating young cells which results in uncontrolled growth and eventually death in susceptible plants due to its chemical resemblance to the auxin hormone [6]. Atrazine is a selective chlorotriazine herbicide and the first triazine herbicide discovered by J.R. Geigy, Ltd., in Switzerland [7]. Atrazine is one of the mostly heavily used in herbicide in Sub-Saharan Africa, an area that is a major maize growing belt on the continent [8]. Atrazine is used on a variety of terrestrial food crops, non-food crops, forests, residential turf, golf course turf, recreational areas, and rangeland [9] and for control of weeds in most farms in Nigeria [8]. Atrazine works by inhibiting photosynthesis by blocking electron transfer at the reducing site (plastoquinone binding site) of photosynthesis complex II in the chloroplasts in higher plants [10, 11]. Although 2,4-D and atrazine based herbicides are among the most commonly used herbicides in many parts of the world (including Nigeria) due to its low cost, selectivity, efficacy and broad spectrum, their negative effects on beneficial and none target organisms are well documented in the literature [12–16].

The environmental fate and effects of 2,4-D on different class of organisms and humans have benefited from recent reviews [17, 18]. 2,4-D based herbicides have been reported to be toxic to skin, eyes, respiratory and gastrointestinal tracts in humans [19] and its usage has been linked to endocrine and cell membranes disruptions, oxidative stress, reproduction toxicity, neurotoxicity, kidney and liver damage and toxicity in humans, fish and amphibians [17, 20, 21]. A wide range of toxic effects in earthworms and beneficial insects have been reported [22, 23]. The honey bee, *Apis mellifera* L. (Hymenoptera: Apidae) colonies fed 2,4-D experienced reduced brood production at a concentration of 100 ppm, and eggs failed to hatch when the colony was exposed to 1000 ppm [24]. Martinez et al. [25] correlated the population declines in three dung beetles with the use of a 2,4-D commercial formulation. Furthermore, commercial formulations of 2,4-D has been reported to be highly toxic, increase development time and decrease male population in larvae of lady beetle, *Coleomegilla maculata* DeGeer (Coleoptera: Coccinellidae) [14].

Despite the plethora of controversies surrounding the toxicity of atrazine based herbicides on non-target organisms such as amphibians [26], they have been proven to have non-target effects on animals [15, 27]. For example, atrazine based herbicides are known to alter

reproductive processes and development in insects, amphibians, fish, reptiles, birds, rodents and goats [15, 16, 28–30] Vogel et al. [15] reported that exposure to atrazine had significant effects on males of *Drosophila melanogaster* Meigen (Diptera: Drosophilidae) mating ability and the number of eggs his partner laid when he was successful at mating. Exposure to atrazine has also been shown to influence gene expression in *Chironomus tentans* Fabricus (Diptera: Chironomidae) [31] and physiology in odonates [32]. In addition, atrazine has been reported to affect diversity in aquatic insect communities [33]. At 2.5 mg atrazine/kg soil, equivalent to 2 kg/ha in the top 10 cm, field and laboratory studies demonstrated that mortality in arthropod collembolids, *Onchiurus apuanicus* was 47% in 60 days [9].

Although many studies have raised questions about the toxicity of 2,4-D and atrazine based herbicides on beneficial and non-target insects such as bees, dung beetles and dragonflies, little or nothing is known about the effects of these herbicides on termite species, despite their ecological, nutritional and ethnopharmacological relevance. Termites (Blattodea) (also known as Blatteria) are undoubtedly key soil organisms in tropical and subtropical soils hence they are called ecosystem engineers (species that modulates the availability of resources for other organisms) or keystone species due to their abundance and impact on ecosystem functions in tropical and sub-tropical ecosystems [34–38]. Although mainly seen as pests of crops, trees and woods, only 185 species of approximately 2800 described termite species, are proven pests [36].

Termites' species such as *Macrotermes* species have considerable influence on the physical, biological and chemical properties of the soil (regulation of soil structure, soil organic matter and nutrient cycling, soil aeration) and consequently water dynamics (improve absorption and storage of water in soil), carbon fluxes and storage, plant growth and overall biodiversity in tropical and subtropical ecosystems [36, 37]. In Africa, *Macrotermes* species are involved in many important ecological processes on savannah, bushlands, and rain forests, and plays a key role in influencing soil dynamics in the tropics [37]. For example, these species of termites act as herbivores and decomposers, feeding on a wide range of living, dead or decaying plant materials [36–39] as well as consumption and turnover of large volumes of soil rich in organic matter and fungi [39]. *Macrotermes* species also recycles dung of primary consumers thereby influencing the functioning of tropical ecosystems [39]. *Macrotermes* species have also been reported to possess entomophagous, ethno-entomological, ethno-medicinal and cultural properties [40].

The hypothesis that the rampant use of 2,4-D and atrazine based herbicides in Nigeria may pose a significant threat to the African mound building termite, *Macrotermes bellicosus* (Smeathman) has to our knowledge not been investigated. It is important to note that, in Nigeria, any possible effects of these herbicides on termites could be aggravated by the users, because many of the rural farmers and/or locals are not educated on the correct use of these herbicides and so seldom apply the correct concentration (O. Uyi, Personal observation). Beneficial insects such worker termites with a high degree of mobility and industry are likely to be caught in direct contact with herbicides at the time of application or immediately after application and may suffer consequences including death as has been documented in other beneficial insects [41, 42]. The goal of this study was to determine the effect of 2,4-D amine salt (Vestamine®) and atrazine (Ultrazine®) based herbicides in *M. bellicosus* by evaluating the mortality and locomotor activity of worker termites after exposure to the recommended concentrations and levels below or above the recommended concentrations. It is important to investigate the effect of herbicides on termites because as ecosystem engineers, their roles in biological, chemical and physical processes in natural and semi-natural ecosystems are far too ecologically relevant to ignore.

## Materials and methods

### Study organism

Several aspects (population ecology, age structure, mound architecture, distribution, density and evolution) of the biology and ecology of *M. bellicosus* which are members of fungus growing sub family of Macrotermitinae and family Termitidae have been studied and documented [43–45]. In semi-arid African environments *Macrotermes* species are among the most dominant in terms of abundance and diversity [46]. *Macrotermes bellicosus* cultivate fungi of the genus *Termitomyces* (Termitomyceteae: Tricholomataceae: Basidiomycota) on special structures within the nest called fungus combs and live in an obligate mutualistic symbiosis with the fungi [47]. They are mostly mound builders and are the largest termite species [34]. The principal food source of *M. bellicosus* consists of dead wood, grass litter, and dung [48]. Outside breaking down dead and decaying plant materials, *Macrotermes* species including *M. bellicosus* acts as ecosystem engineers by changing the surrounding environment and creating conditions different from adjacent soil through structuring and controlling to a large extent, the flows of energy and matter in the tropics and subtropics through allogenic engineering processes [34, 36, 37].

### Collection of termites

*Macrotermes bellicosus* worker termites were collected from a termite mound in a field within the vicinity of Faculty of Life Sciences (6˚ 23' 55"N, 5˚ 36' 54"E, University of Benin, Benin City, Nigeria. Since the establishment of the Faculty in June 2005, mechanical control is and remains the only method of weed control on the property. Therefore, the termites have never been directly exposed to herbicides. The Dean of the Faculty of Life Sciences, University of Benin, Benin City, granted us permission to use the field site within the faculty. The termite mound was dug up using a shovel and *M. bellicosus* were collected with a mix of soil and these (soil and termite) where placed in a plastic container (25 Litres) and taken to the laboratory of the Department of Animal and Environmental Biology and kept in a cool dark place until needed for the experiment (not more than 6 hours). When needed the termites were removed from the soil using a fine soft camel brush. Both mortality and locomotion performance tests were conducted in the laboratory of the Department of Animal and Environmental Biology, University of Benin, Benin City, Nigeria between May and August 2018.

### Herbicides

Vestamine[R] (2,4-D amine salt) and Ultrazine[R] (Atrazine 80% wp) which are among the commonly used herbicides for controlling weeds in Benin City and other parts of Nigeria, were chosen for this study. The recommended concentrations of 2,4-D and atrazine based herbicides (2.5 litres per 200 litres of water = 6.25 ml per 500 ml of water for 2,4-D and 120g per 16 litres of water = 3.75 g per 500 ml of water for atrazine) were obtained and dissolved in 500 ml of water. The mixture was shaken thoroughly to ensure an even suspension. Vestamine[R], a commercially formulated 2,4-D containing 720 g/l w/v 2,4-D as the Amine Salt (600 g/l as extractable acid) (Nanjing CF Agrochemicals Co Ltd, Honghanyo, Luhe District, Nanjing, China) was used at the concentration recommended (6.25 ml per 500 ml of water) by the manufacturer and at 0.25- and 0.5-fold (1.563 ml per 500 ml of water, 3.125 ml per 500 ml of water, respectively) of the recommended concentration. Ultrazine[R], a commercially formulated Atrazine containing 80% wp atrazine in powdered form (Zhejiang Zhongshan Chemical Industry Group Co., Ltd. Zhongshan, Zhejiang, China) was used at the concentration recommended (3.75 g per 500 ml of water) by the manufacturer and at 0.25-, 0.5-, 2- and 4-fold

(0.938 g per 500 ml of water, 1.875 g per 500 ml of water, 7.5 g per 500 ml of water and 15 g ml per 500 ml of water) of the recommended concentration. Distilled water was used as the control. The higher concentrations of atrazine used in this study were similar to what locals and rural farmers commonly used in southern Nigeria (O. Uyi, Personal observation) due to little or no education on the use atrazine based herbicide. All tests were conducted using freshly prepared solutions in distilled water at $25 \pm 2$°C, $65 \pm 10$% RH.

## Mortality bioassay

Worker termites were sorted into groups of 5 and placed in small Petri-dishes (90-mm-diameter) lined with a disc of filter paper (Whatman No. 1) moistened with 0.6 ml of water. Fresh dilutions of test chemicals were mixed less than 1 hour before application. A 30 μl droplet of the test chemicals at different concentrations (1.563, 3.125, 6.25ml per 500 ml of water for Vestamine® and 0.938, 1.875, 3.75, 7.5 and 15.0 g per 500 ml of water for Ultrazine®) was applied dorsally with a syringe to each insect. Since worker termites are the caste of termites which go out often to forage, they are likely to come in direct contact with these herbicides at the time of, or post- application because of the frequency of usage by locals and farmers. A chemical-free control was set up whereby a 30 μl droplet of distilled water was applied to each insect. Ten replicates of five insects were used for each of the herbicide treatment and control group. A total of 200 worker termites were used for Vestamine® (5 worker termites x 4 herbicide treatments x 10 replicates = 200) while a total of 300 worker termites were used for Ultrazine® (5 worker termites x 6 herbicide treatments x 10 replicates = 300). All Petri dishes having termites (treated with the test herbicides or untreated) were kept in darkness by covering with a black plastic sheet to simulate the dark environment of the termite mounds at a temperature of $25 \pm 2$°C and relative humidity of $65 \pm 10$%. Insect mortality was monitored every 12 hours for up to 24 hours after application of the herbicides. A worker termite was regarded as dead if it showed no signs of movement when touched lightly with a soft camel hair brush. Mortality was recorded and percentage mortality was calculated. Individual termites were used as the level of replication for both the tests of significance for mortality and for the $LC_{50}$ and $LC_{90}$ analysis.

## Locomotion performance trial

During spraying or application of atrazine or 2,4-D based herbicides by locals or farmers, foraging worker termites might be sprayed upon or come in contact with residues of these herbicides and this might have an effect on the locomotion ability of this species. *Macrotermes bellicosus* workers were obtained for the locomotion trials as described above on the day the trial was performed. After collection, the insects were placed in Petri dishes in preparation for the experiment. The effect of herbicides on termite's locomotor ability was determined by recording the speed of movement and distance covered in 30 seconds when treated with a range of concentrations of the test herbicides (1.563, 3.125, 6.25 ml per 500 ml of water for Vestamine® and 0.938, 1.875, 3.75, 7.5 and 15 g per 500ml of water for Ultrazine®). Five termite workers were exposed individually to a particular concentration of the test chemicals by applying a 30 μl droplet of the different herbicide treatments dorsally to the insect and kept in an open Petri dish for 2 minutes to acclimatize after which the insect was allowed to walk on a stage of 90 x 60 cm (plyboard wrapped with a white cupboard paper) for 15 seconds. An HB pencil was used to trace the distance travelled by the insect. Because the insect travelled in a zigzag manner after treatment with herbicides, a thread was used to measure the distance walked by the insect and the length of the thread was measured with a measuring tape and recorded. A total of 10 replicates were used for each concentration of the test herbicides. A

total of 200 worker termites were used for Vestamine® (5 worker termites x 4 herbicide treatments x 10 replicates = 200) while a total of 300 worker termites were used for Ultrazine® (5 worker termites x 6 herbicide treatments x 10 replicates = 300). The speed of individual termite was calculated by dividing the distance covered (in cm) by the total time the termite was exposed for (15 seconds).

## Statistical analysis

Following arcsine square root transformation of the mortality data, the effects of different concentrations of herbicides (Vestamine® and Ultrazine®) and exposure time on mortality was analyzed using a Generalized Linear Model (GLZ) (assuming normal distribution with an identity link function). When the overall results were significant in the GLZ analysis, the difference among the treatments was compared using the sequential Bonferroni test. Probit regression was used to estimate the concentrations of both herbicides estimated to cause 50 and 90% mortality ($LC_{50}$ and $LC_{90}$); the concentrations causing 50% and 90% of tested individuals to die in a given period (i.e. 12 and 24 hours). The effect of herbicides on the speed of *M. bellicosus* was evaluated using General Linear Model Analysis of Variance (GLM ANOVA). When the overall results were significant in the GLM ANOVA, the difference among the treatments was compared using Tukey's Honest Significant Difference (HSD) test. All analyses were performed using SPSS Statistical software, version 20.0 (IBM SPSS, Chicago, IL, USA).

## Results

### Mortality bioassay

Vestamine® (2, 4-D based herbicide) and Ultrazine® (atrazine based herbicide) caused varying and significant levels of mortality against *M. bellicosus* (Figs 1 and 2). Mortality caused by Vestamine® varied as a function of concentration (GLZ: Wald $\chi^2_3$ = 1266.00; P = 0.0001) and exposure time (Wald $\chi^2_1$ = 81.50; P = 0.0001) (Fig 1). The interaction between concentration and time also varied significantly (Wald $\chi^2_3$ = 77.50; P = 0.0001). Higher mortality of worker termites were recorded at higher concentrations, which increased over time until 24 hours where 100% mortality occurred in the highest treatment (the recommended concentration). The recommended concentration caused 100% mortality in treated termites after 12 and 24 hour exposure. The 0.25- and 0.5-fold below the recommended concentration of Vestamine® caused 44 to 72% mortality and 88 to 96% mortality after 12 and 24 hours, respectively (Fig 1). Percentage mortality caused by Ultrazine® was influenced by concentrations (Wald $\chi^2_3$ = 846.67; P = 0.0001), but not exposure time (Wald $\chi^2_1$ = 7.33; P = 0.067) and their interactions (Wald $\chi^2_5$ = 4.67; P = 0.458) (Fig 2). The recommended concentration respectively caused 92 and 100% mortality in treated termites after 12 and 24 hour exposure. The 0.25-, 0.5-, 2-, and 4-fold below the recommended concentration of atrazine caused 80 to 100% mortality and 96 to 100% mortality after 12 and 24 hours, respectively (Fig 2). Concentrations of Vestamine® and Ultrazine® estimated to cause 50% ($LC_{50}$) and 90% ($LC_{90}$) mortality in termites were calculated based on the mortality results (Table 1). $LC_{50}$ and $LC_{90}$ decreased with increased exposure time for both herbicides. Following 24-hour exposure to Vestamine®, $LC_{50}$ and $LC_{90}$ were 1.028 and 2.031 ml per 500 ml of water. The calculated $LC_{50}$ and $LC_{90}$ for Ultrazine® were 0.212 and 1.348 ml per 500 ml of water.

### Locomotion performance

Locomotion performance of *M. bellicosus* varied as a function of herbicide concentrations. Overall locomotor performance reduced with increase in concentration irrespective of

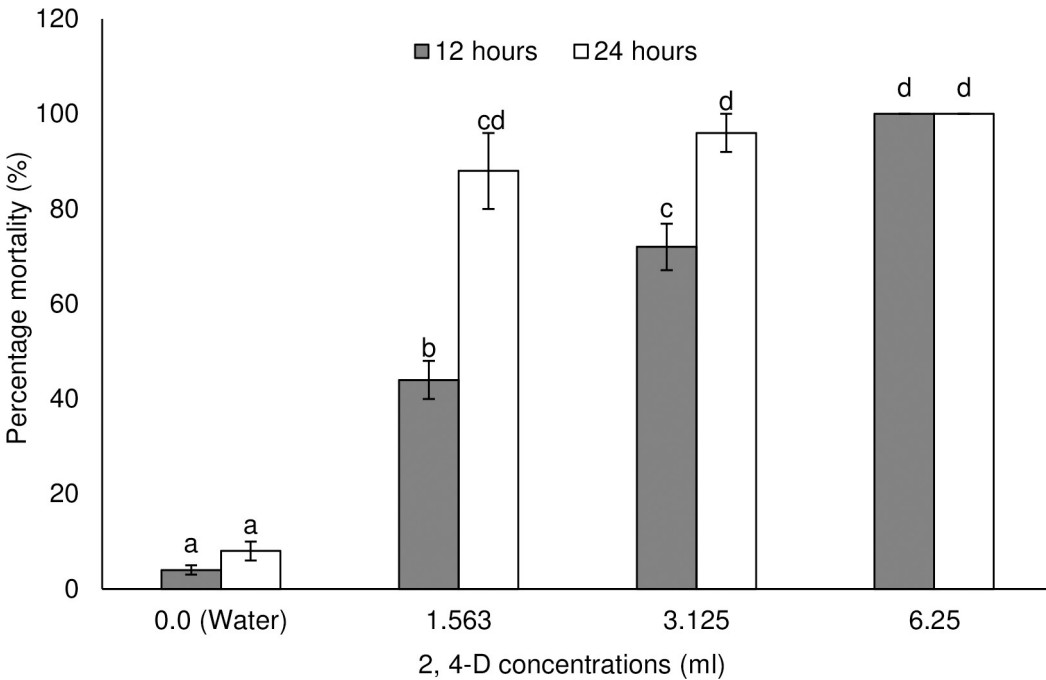

**Fig 1. Percentage mean (±SE) mortality of *Macrotermes bellicosus* following exposure to different concentrations of Vestamine® for 12 and 24 hours.** Means capped with different letters are significantly different (sequential Bonferroni test, P<0.05).

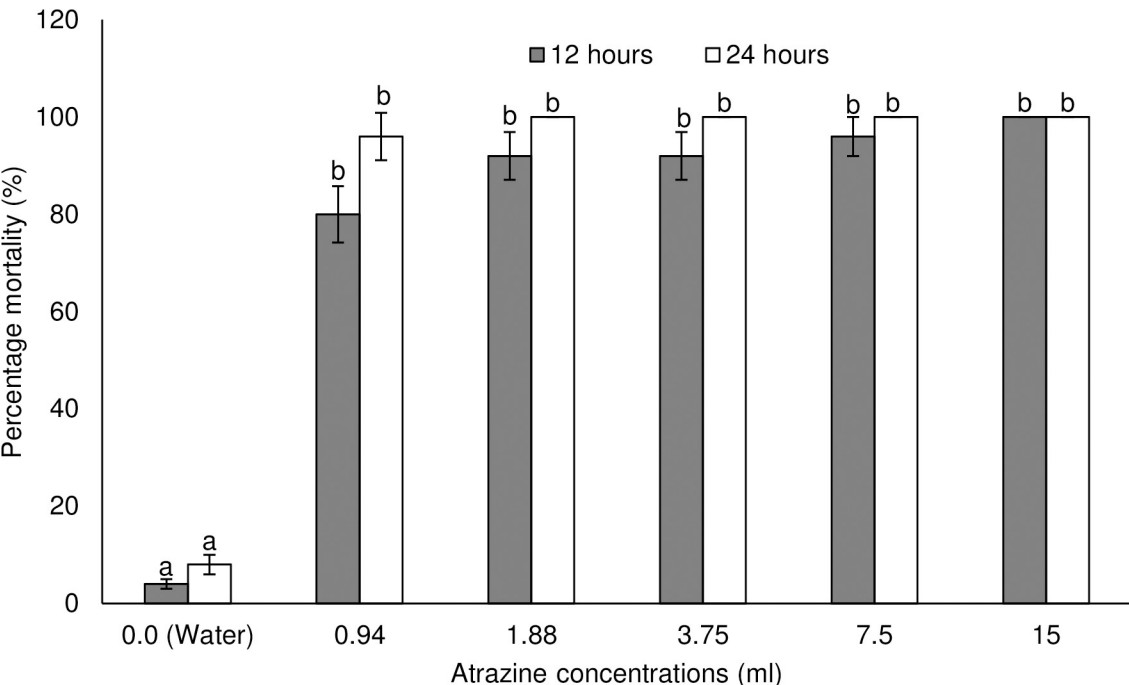

**Fig 2. Percentage mean (±SE) mortality of *Macrotermes bellicosus* following exposure to different concentrations of Ultrazine® for 12 and 24 hours.** Means capped with same letters are not significantly different (sequential Bonferroni test, P>0.05).

**Table 1. Index of toxicity ($LC_{50}$ and $LC_{90}$) of *Macrotermes bellicosus* when exposed to different concentrations of Vestamine® and Ultrazine® for 12 and 24 hours.**

| Herbicide | Exposure time (hours) | Index of toxicity | | 95% confidence interval | |
|---|---|---|---|---|---|
| | | $LC_{50}$ (ml) | $LC_{90}$ (ml) | $LC_{50}$ (ml) | $LC_{90}$ (ml) |
| **Vestamine®** | 12 | 2.091 | 3.911 | 1.512–3.144 | 2.782–5.432 |
| | 24 | 1.028 | 2.031 | 0.778–1.468 | 1.532–4.742 |
| **Ultrazine®** | 12 | 0.571 | 2.515 | 0.282–0.763 | 2.941–4.111 |
| | 24 | 0.212 | 1.348 | 0.183–0.279 | 1.022–2.583 |

herbicide tested. The running speed of termites varied significantly ($F_{3,39}$ = 137.41; P = 0.0001) with termites treated with water (control treatment) running the highest speed (3.56 cm/s) compared to those that were treated with different concentrations of Vestamine® (Fig 3). The running speed of termites exposed to Ultrazine® also differed ($F_{5,59}$ = 81.65; P = 0.0001) with workers exposed to the recommended concentration running significantly slower (1.88 cm/s) compared with the control (treated with water) group that had the fastest speed (3.81 cm/s) (Fig 4).

## Discussion

The increasing use of herbicides in natural and semi natural ecosystems (e.g. agroecosystems) over the past years have led to some deleterious effects on some biotic components of the eco-system including non-target and beneficial organisms and even human health [18, 49]. Until recently, the adverse effects of herbicides on non-target or beneficial organisms have not received much attention in Nigeria [5] and toxicological studies of herbicides on termites are still scarce or non-existent. In this study, the recommended concentrations (by manufacturers) of 2,4-D amine salt (Vestamine®) and atrazine (Ultrazine®) based herbicides and values below and above the recommended concentrations were applied on termites. The results of

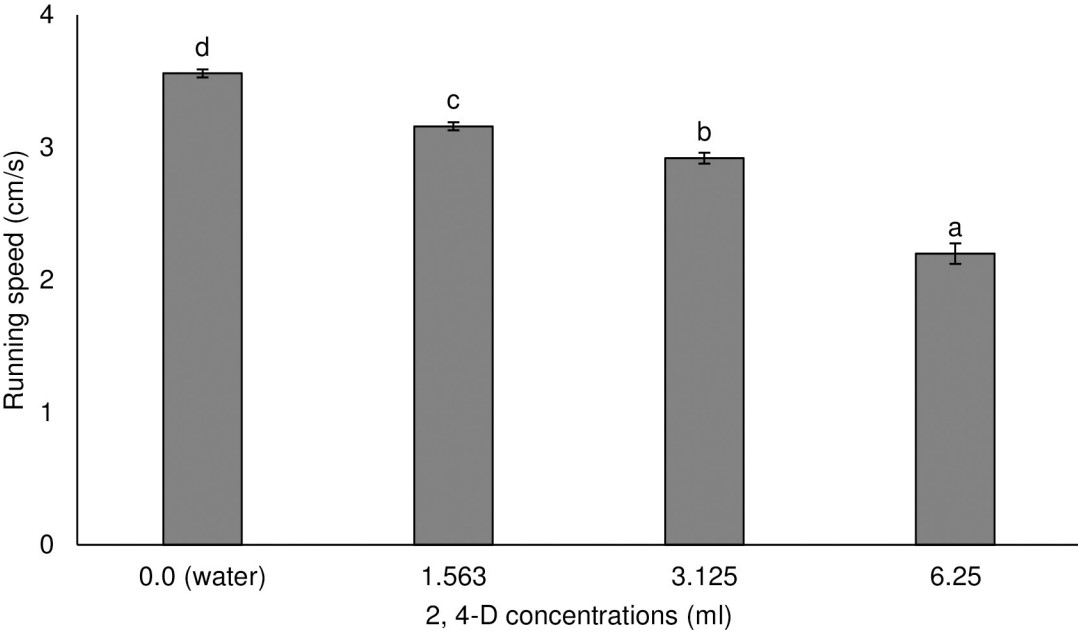

**Fig 3. Mean (±SE) running speed of *Macrotermes bellicosus* after exposure to different concentrations of Vestamine®.** Means capped with different letters are significantly different [Tukey's Honest Significant Difference (HSD) test: P<0.05].

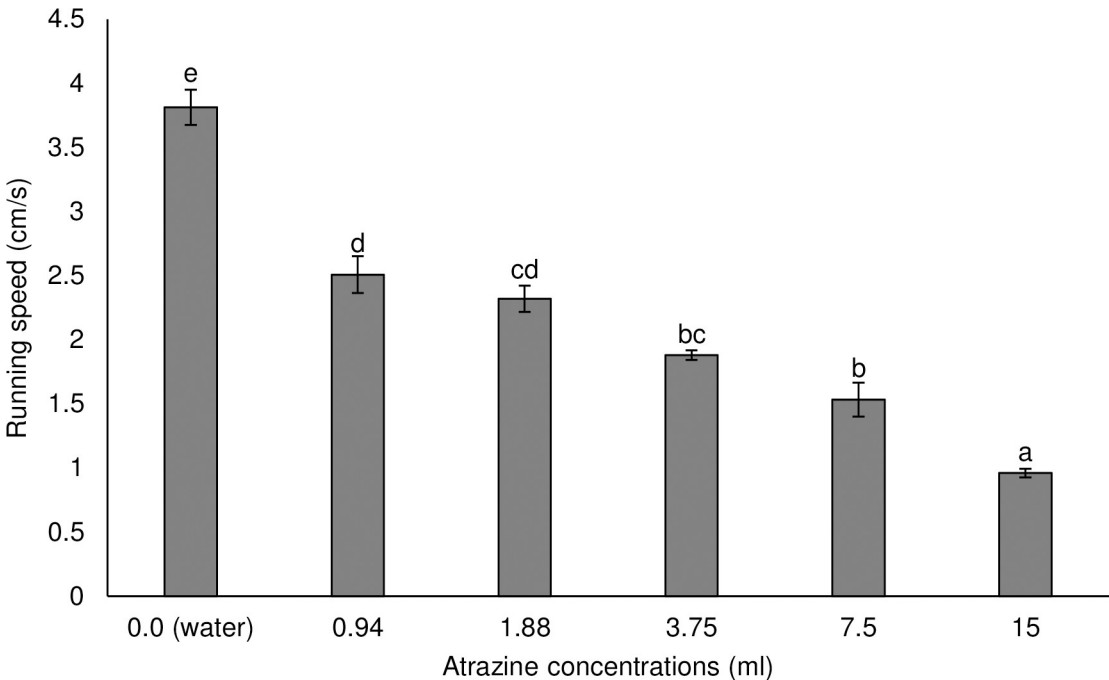

**Fig 4. Mean (±SE) running speed of *Macrotermes bellicosus* after exposure to different concentrations of Ultrazine®.** Means capped with different letters are significantly different [Tukey's Honest Significant Difference (HSD) test: P<0.05].

this study demonstrates that the recommended concentrations of Vestamine® and Ultrazine® caused high mortality (100% after 24 hours exposure) and impaired locomotion ability in workers of the African mound building termite, *M. bellicosus*.

Exposure to Vestamine® (2, 4-D based herbicide) caused significant mortality in termites but this was dependent on exposure duration and concentrations tested. Concentrations of Vestamine® estimated to cause 50% and 90% mortality ($LC_{50}$ and $LC_{90}$) were generally higher after 12-hour exposure than 24-hour exposure implying that toxicity increased with time. $LC_{50}$ and $LC_{90}$ were also higher after 12-hour exposure than 24-hour exposure to Ultrazine®. Based on the $LC_{50}$ values and the recommended concentration for use, Vestamine® and Ultrazine® were highly toxic to termites as the $LC_{50}$ and $LC_{90}$ fell below the recommended dosage range at both 12 and 24 hours. This study showed that the manufacturer's recommended concentration of Vestamine® and Ultrazine® caused 100% mortality in worker termites at 24 hours after treatment. The significant interactions in percentage mortality between concentration and exposure time (12 and 24 hours) suggests that the effect of Vestamine® was more apparent or stronger in the 24- hour exposure trial compared to the 12-hour exposure trial, but the biological significance of this remains to be seen. While a few authors have reported the safety of 2, 4-D based herbicides on beneficial insects [50], studies reporting the toxicity of 2, 4-D based herbicides are not uncommon. For example Hill et al. [51] reported that the recommended concentration of a commercially formulated 2,4-D caused up to 80% mortality in the mirid, *Ecritotarsus catarinensis* (Carvalho) (Hemiptera: Miridae) after the insect was exposed for 72 hours. In addition, Freydier and Lundgren [14] reported 80% mortality in ladybug, *Coleomegilla maculata* De Geer (Coleoptera: Coccinelidae) larvae due to exposure to 2,4-D. Furthermore, Adam [52] reported four times mortality in three species of coccinelid beetle larvae, *Coccinella transversoguttata* (Fald.), *Hippodamia tredecimpunctata* (L.) and *Coccinella perplexa* (Muls.) in six different age groups sprayed with 2,4-D compared to the control (water).

In our study, the highest mortality was recorded at the highest concentrations and longest exposure duration, suggesting time and dose-dependent survival and concentration graded lethality. In contrast, mortality caused by Ultrazine® was high irrespective of concentration and exposure time, which may be due to the high toxicity of atrazine based herbicides as has been previously reported [53]. Godfrey and Rypstra [53] reported significant reduction in adult lifespan of agrobiont wolf spider *Pardosa milvina* on exposure to atrazine based herbicide.

Previous studies and reviews have reported mortality on exposure to either atrazine or 2,4-D based herbicides (either directly or indirectly by causing life threatening impacts) in humans [54–56]. Atrazine and 2,4-D have been shown to increase insecticide toxicity in southern armyworm *Spodoptera eridania* (Stoll) (Lepidoptera: Noctuidae) [57]. The high percentage mortality caused by the recommended concentrations of Vestamine® and Ultrazine® in worker termites recorded in this study suggests that exposure to these herbicides and other related chemicals might lead to a reduced work force which may result in a reduction or loss of their ecosystem engineering functions and starvation of the colony as other castes depend on the food and water brought back by foraging workers. It is not impossible that termites that take up herbicides or survive exposure and are able to make their way back to the colony may take some of the residues of these herbicides back and this may lead to a buildup of pesticide toxicity in the mound. For example, atrazine residues have been found in nearly 14% of North American bee colonies including brood combs where bees lay eggs and larvae develop [58] and has been found in high concentrations in honeybee colonies that were either sick or dead [59].

The effect of exposure to pesticides on locomotion of beneficial arthropods is often indirectly studied [60]. In this study, the recommended concentration of Vestamine® (2, 4-D based herbicide) reduced the running speed by 38% in treated worker termites compared to the control treatment (water). Similarly, the recommended concentration of Ultrazine® (atrazine based herbicide) reduced the running speed by 51% in treated worker termites compared to the control treatment (water). The reduced mobility (reduced running speed) in worker termites might be due to direct intoxication resulting in knock-down effect, trembling and tumbling or rotating [60]. Exposure to low concentrations of the herbicides resulted in an increased mobility of worker termites and conversely exposure to higher concentrations reduced movement. The reduced mobility in the worker termites may limit the ability of this caste to transport food to the colony and perform numerous ecological functions. Also, the reduced mobility in worker termites may increase their vulnerability to predation and harsh environmental conditions as has been reported in other insects [61]. Although studies on the effect of 2,4-D and atrazine based herbicides on the locomotion performance in insects are still scarce, there are few available literatures on glyphosate and bees, with contrasting results [61,63]. For example Herbert et al. [62] reported no effect on the locomotive ability of forager honey bees, *Apis mellifera* L. (Hymenoptera: Apidae) when foragers collected sucrose contaminated with glyphosate at an artificial feeder but Balbuena et al. [63] reported that exposure to glyphosate had an effect on honeybee navigation. In other taxa, organochlorine herbicides (endosulfan and chlordane) have been reported to decrease swim speed and activity levels in both fish and amphibians [64].

Beyond the fact that the recommended concentrations of both Vestamine® (2, 4-D based herbicide) and Ultrazine® (atrazine based herbicide) resulted in 100% mortality and significantly reduced mobility in worker termites, one of the interesting findings of the study was that concentrations of both herbicides far below the recommended rates caused 100% mortality in the studied species, suggesting that even the littlest exposure of termites to these chemicals can reduce their population. Our findings therefore reinforce the warning that herbicides,

although important for agriculture, needs to be used more cautiously, applied only in the indicated amounts and, as far as possible, replaced by other methods that are less harmful to the environment and biodiversity. Understanding the mechanism of toxicity is key to fully elucidate a toxicant's potential impacts and should be the focus of future studies. Furthermore, experimental studies in the field are rare, and most of the studies we cited here were conducted in the laboratory. Future comparative field and laboratory research should be prioritized as this will elucidate the complexity in behavioral responses of animals to herbicides. Also, future studies should aim to provide more ecological relevance to these findings by investigating the effect of herbicides on related behavioural measures such as termite navigation. In sum, we presented data for the first time to show that Vestamine® (2, 4-D based herbicide) and Ultrazine® (atrazine based herbicide) are highly toxic and are capable of causing high mortality and reduced mobility in workers of the African mound building termite, *M. bellicosus* and can negatively affect the ecosystem services provided by these species which may compromise the integrity of the ecosystem. Therefore, these herbicides should be used by skilled personnel and we recommend that locals and farmers should be educated on the proper use of 2.4-D and atrazine based herbicides due to their lethal and sub-lethal effects.

## Acknowledgments

We thank two anonymous reviewers for their constructive comments on early drafts and Mr. Joshua Mfom Friday, for assistance with collecting the termites used for this study.

## Author Contributions

**Conceptualization:** Afure J. Ejomah, Osariyekemwen O. Uyi.

**Data curation:** Afure J. Ejomah, Osariyekemwen O. Uyi.

**Formal analysis:** Afure J. Ejomah, Osariyekemwen O. Uyi.

**Funding acquisition:** Osariyekemwen O. Uyi.

**Investigation:** Afure J. Ejomah, Osariyekemwen O. Uyi.

**Methodology:** Afure J. Ejomah, Osariyekemwen O. Uyi.

**Project administration:** Osariyekemwen O. Uyi, Sese-Owei Ekaye.

**Resources:** Osariyekemwen O. Uyi, Sese-Owei Ekaye.

**Supervision:** Osariyekemwen O. Uyi, Sese-Owei Ekaye.

**Validation:** Afure J. Ejomah, Osariyekemwen O. Uyi, Sese-Owei Ekaye.

**Visualization:** Afure J. Ejomah, Osariyekemwen O. Uyi, Sese-Owei Ekaye.

**Writing – original draft:** Afure J. Ejomah, Osariyekemwen O. Uyi, Sese-Owei Ekaye.

**Writing – review & editing:** Afure J. Ejomah, Osariyekemwen O. Uyi, Sese-Owei Ekaye.

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
