## [Decision Letter · Decision Letter 0]

18 Dec 2019

PONE-D-19-19112

Exposure of the African mound building termite, Marcrotermes bellicosus workers to 2,4-D and atrazine based herbicides caused high mortality and impaired locomotor response

PLOS ONE

Dear Dr Uyi,

Thank you for submitting your manuscript to PLOS ONE. After careful consideration, we feel that it has merit but does not fully meet PLOS ONE’s publication criteria as it currently stands. Therefore, we invite you to submit a revised version of the manuscript that addresses the points raised during the review process.

Both reviewers commented on the length of the discussion and suggested that it could be reduced without detracting from the manuscript. Please pay particular attention to the comments from Reviewer 1 regarding the relevance of the testing procedure to expected exposure/application routes as well as discussion of potential confounding effects from other chemicals present in the pesticide formulations. Prepare a careful point-by-point response for all comments from both reviewers.

We would appreciate receiving your revised manuscript by Feb 01 2020 11:59PM. To enhance the reproducibility of your results, we recommend that if applicable you deposit your laboratory protocols in protocols.io, where a protocol can be assigned its own identifier (DOI) such that it can be cited independently in the future. For instructions see: http://journals.plos.org/plosone/s/submission-guidelines#loc-laboratory-protocols

We look forward to receiving your revised manuscript.

Kind regards,

Carla A Ng

Academic Editor

PLOS ONE

Journal Requirements:

1. In your Methods section, please provide additional location information of the collection site, including geographic coordinates for the data set if available.

Reviewers' comments:

Reviewer's Responses to Questions

**Comments to the Author**

1. Is the manuscript technically sound, and do the data support the conclusions?

Reviewer #1: Partly

Reviewer #2: Yes

2. Has the statistical analysis been performed appropriately and rigorously? 

Reviewer #1: I Don't Know

Reviewer #2: Yes

3. Have the authors made all data underlying the findings in their manuscript fully available?

Reviewer #1: No

Reviewer #2: Yes

4. Is the manuscript presented in an intelligible fashion and written in standard English?

Reviewer #1: Yes

Reviewer #2: Yes

5. Review Comments to the Author

Reviewer #1: Review of PONE-D-19_19112

In this manuscript the authors investigate the effects of two herbicides on mortality and locomotion in an African mound building termite. The authors investigate several concentrations that are potentially relevant to those concentrations at the time of application. Both herbicides has significant detrimental effects on survival and locomotion at all the concentrations tested. I certainly think that understanding how herbicides can affect non-target organisms is an important part of understanding how humans are affect ecosystems.

Comments:

1. The running speed analysis is completely redundant with the distance covered. Only one should be included.

2. I am a bit concerned with the application procedure. The authors apply 30 ul of a pretty high concentration of herbicide directly to the termite. I don’t have any feel for whether that is relevant. I would like the authors to demonstrate that 30 ul directly on a termite is relevant under “realistic” application conditions.

3. I am a bit confused about the actual stats. It seems that for the tests of significance for mortality the petri plate was the level of replicate but then for the LC50 and 90 analysis the individual was the level of replication. Could this be explained a bit more clearly.

4. I am concerned with estimates of LC50 for the atrazine. This first measurement, for the lowest concentration, is 80% mortality and thus it seems difficult to get an accurate estimate of LC50.

5. Line 259. The authors state there is a time by concentration interaction. If so, effect of concentration should be analyzed separately for each time and the interpretation of main effects is difficult when an interaction exists. Furthermore, what is the biological relevance of this interaction?

6. The authors focus on discussions of atrazine and 2,4 D when they are really testing the effects of two commercial herbicides that contain these chemicals. Water is not the best control. It would be better to have tested only the effects of atrazine or 2,4 D in isolation or had controls that contained all the other compounds (solvents and such) present in the commercial formulations. For example DMSO is sometimes a solvent and is known to affect fruit flies. This needs to be specifically addressed as a major caveat and the description of the effects being due to atrazine or 2.4 D should be tempered

Smaller comments:

7. Collection of termites: When was the “Faculty of Life Sciences” established. This is relevant because the authors state no herbicides have been applied to the locality since then.

8. Line 181. I assume the authors mean suspension rather than solution. I don’t know the exacts, but many commercial formulations of herbicides are suspensions and must be constantly mixed during application.

9. The discussion is too long. The section starting with line 327 can be dramatically shortened.

Reviewer #2: I have inserted my comments through comments, please incorporate all the suggested changes. Once you incorporate the comments, then this article will be accepted. Discussion part is way longer, try to be concise and condense based results. Thanks

6. PLOS authors have the option to publish the peer review history of their article (what does this mean?). If published, this will include your full peer review and any attached files.

Reviewer #1: No

Reviewer #2: No

---

## [Author Response · Author response to Decision Letter 0]

2 Jan 2020

Rebuttal Letter: Response (in red fonts) to the Academic Editor and Reviewers (Manuscript number: PONE-D-19-19112). 

Thank you for submitting your manuscript to PLOS ONE. After careful consideration, we feel that it has merit but does not fully meet PLOS ONE’s publication criteria as it currently stands. Therefore, we invite you to submit a revised version of the manuscript that addresses the points raised during the review process.

Many thanks for this

Academic Editor:

1. Both reviewers commented on the length of the discussion and suggested that it could be reduced without detracting from the manuscript. 

Attended to. We have now shortened this discussion section

2. Please pay particular attention to the comments from Reviewer 1 regarding the relevance of the testing procedure to expected exposure/application routes as well as discussion of potential confounding effects from other chemicals present in the pesticide formulations. 

We have now attended to this below.

We think that the application of 30 µl directly on individual termites is possible under realistic conditions because a preliminary observation on a property that was sprayed with herbicide in January 2017 showed that the ant, Crematogaster africana and the African mound termite received significant droplets and their attempts to escape led to a further uptake of the sprayed herbicide from the sprayed plant materials. This observation partly influenced the study presented in our manuscript. Hill et al. (2012) dorsally applied 20 µl droplet of herbicides to the weevils Eccritotarsus catarinensis and Neochetina eichhorniae. These insects are way smaller than the worker termites used in this study. Therefore, we think 30 µl is not too much because termite can easily pick up more than this droplet size in nature. It is important to note here that, in Nigeria, many of the rural farmers (peasants) and/or locals are not educated on the correct use of these herbicides and so seldom apply the correct concentration or application conditions. Therefore in such a situation, the termites will potentially receive higher droplets and concentrations of herbicides. Finally, the concentrations of the herbicides tested were not pretty high as we only used 2 levels below the recommended concentrations and the recommended concentration for Vestamine®. For Ultrazine®, we used 2 levels below the recommended concentration, the recommended concentrations and 2 level above the recommended concentration. 

We clearly understand and appreciate the concern of Reviewer 1 on the use of water as control in our experiment. The use of water as a control in this kind of experiment where the effect of commercially formulated herbicides on beneficial insects are not uncommon (e.g. Hill et al., 2012, Biocontrol Science and Technology; Abraham et al., 2018, Entomologia Experimentalis et Applicata). For example, Hill et al. (2102) and Abraham et al. (2018) used water as control in their test of the effect of some commercially formulated herbicides on bees and some weevils (biocontrol agents of the invasive water hyacinth in South Africa). Therefore, we are confident that our usage of water as control is not aberration.

3. Prepare a careful point-by-point response for all comments from both reviewers.

Attended to

Journal Requirements:

Attended to

1. In your Methods section, please provide additional location information of the collection site, including geographic coordinates for the data set if available.

Done (see Materials and methods section)

Done (see Materials and methods section) 

Reviewer #1: PONE-D-19_19112

In this manuscript the authors investigate the effects of two herbicides on mortality and locomotion in an African mound building termite. The authors investigate several concentrations that are potentially relevant to those concentrations at the time of application. Both herbicides has significant detrimental effects on survival and locomotion at all the concentrations tested. I certainly think that understanding how herbicides can affect non-

target organisms is an important part of understanding how humans are affect ecosystems.

Many thanks for your comments. We totally agree with your submissions.

Comments: 

1. The running speed analysis is completely redundant with the distance covered. Only one should be included.

We have now deleted the graphs of the distance covered for both herbicides and elected to retain the running speed as advised by Reviewer 1. We have deleted all parts of the manuscript that reported or discussed “distance covered” by termites in both the Results and Discussion sections. This has partly shortened the length of the manuscript. 

2. I am a bit concerned with the application procedure. The authors apply 30 ul of a pretty high concentration of herbicide directly to the termite. I don’t have any feel for whether that is relevant. I would like the authors to demonstrate that 30 ul directly on a termite is relevant under “realistic” application conditions.

We think that the application of 30 µl directly on individual termites is possible under realistic conditions because a preliminary observation on a property that was sprayed with herbicide in January 2017 showed that the ant, Crematogaster africana and the African mound termite received significant droplets and their attempts to escape led to a further uptake of the sprayed herbicide from the sprayed plant materials. This observation partly influenced the study presented in our manuscript. Hill et al. (2012) dorsally applied 20 µl droplet of herbicides to the weevils Eccritotarsus catarinensis and Neochetina eichhorniae. These insects are way smaller than the worker termites used in this study. Therefore, we think 30 µl is not too much because termite can easily pick up more than this droplet size in nature. It is important to note here that, in Nigeria, many of the rural farmers (peasants) and/or locals are not educated on the correct use of these herbicides and so seldom apply the correct concentration or application conditions. Therefore in such a situation, the termites will potentially receive higher droplets and concentrations of herbicides. Finally, the concentrations of the herbicides tested were not pretty high as we only used 2 levels below the recommended concentrations and the recommended concentration for Vestamine®. For Ultrazine®, we used 2 levels below the recommended concentration, the recommended concentrations and 2 level above the recommended concentration. 

3. I am a bit confused about the actual stats. It seems that for the tests of significance for mortality the petri plate was the level of replicate but then for the LC50 and 90 analysis the individual was the level of replication. Could this be explained a bit more clearly.

Many thanks for these observations as they have provided us with the opportunity to clarify things. We have now include the following statement in the revised manuscript “The individual was the level of replication for both the tests of significance for mortality and for the LC50 and 90 analysis”. 

It is not apparent in in the stats degrees of freedom in the tests of significance for mortality because we performed a Generalized Linear Model (GLZ) analysis where only the degrees of freedom for treatment are shown as a subscript beneath the Chi Square values (e.g. GLZ: Wald χ23=……). A normal ANOVA would have allowed us to write out the treatment and error/total degrees of freedom in a more orthodox sense (F3,199=…). Just to clarify things once more, ten replicates of five insects were used for each of the herbicide treatment and control group. A total of 200 worker termites were used for Vestamine® (5 worker termites x 4 herbicide treatments x 10 replicates = 200) while a total of 300 worker termites were used for Ultrazine® (5 worker termites x 6 herbicide treatments x 10 replicates = 300) based herbicide.

4. I am concerned with estimates of LC50 for the atrazine. This first measurement, for the lowest concentration, is 80% mortality and thus it seems difficult to get an accurate estimate of LC50. 

We had good number of replicates and we are confident that the estimate of the LC50 and LC90 is accurate as per Lopez et al. (2015, Journal of Economic Entomology). In any case we have confidence intervals for the calculated index of toxicity.

5. Line 259. The authors state there is a time by concentration interaction. If so, effect of concentration should be analyzed separately for each time and the interpretation of main effects is difficult when an interaction exists. Furthermore, what is the biological relevance of this interaction?

Many thanks for your comments. The statistics we performed is straightforward: the effect of concentrations of herbicides (factor 1) and exposure time (factor 2) on mortality was analysed using GLZ analysis. Interpreting this kind of analysis (2 factor analysis) s pretty straightforward and widely accepted (Lukowski et al., 2015; PLoS ONE; Uyi et al., 2018, PLoSONE,). We have elected not to separate it because we feel the current analysis is adequate. Moreover, the Associate Editor and Reviewer 2 are comfortable with it. 

Following the advice of Reviewer 1, we have now interpreted the interaction between concentration and exposure time and inserted the line below in our discussion section (Lines 337-340 in the revised manuscript):

“The significant interactions in percentage mortality between concentration and exposure time (12 and 24 hours) suggests that the effect of Vestamine® was more apparent or stronger in the 24- hour exposure trial compared to the 12-hour exposure trial, but the biological significant of this remains to be seen”. 

6. The authors focus on discussions of atrazine and 2,4 D when they are really testing the effects of two commercial herbicides that contain these chemicals. Water is not the best control. It would be better to have tested only the effects of atrazine or 2,4 D in isolation or had controls that contained all the other compounds (solvents and such) present in the commercial formulations. For example DMSO is sometimes a solvent and is known to affect fruit flies. This needs to be specifically addressed as a major caveat and the description of the effects being due to atrazine or 2.4 D should be tempered.

We have now reverted to using the two commercially formulated herbicides (Vestamine® and Ultrazine®) tested. We have replaced 2,4-D and Atrazine with Vestamine® and Ultrazine® in relevant parts or portions of the manuscript. Our manuscript is on the effects of Vestamine® and Ultrazine® on termites. We did not set out to test the effect of atrazine and 2,4-D salt on termites. We only used Vestamine® and Ultrazine® as proxy for 2,4-D and atrazine. We have now focused our manuscript on Vestamine® and Ultrazine®. 

Based on the above, we have now changed the title of the manuscript to reflect “commercially formulated”. The revised title reads “Exposure of the African mound building termite, Marcrotermes bellicosus workers to commercially formulated 2,4-D and atrazine caused high mortality and impaired locomotor response”.

We clearly understand and appreciate the concern of Reviewer 1 on the use of water as control in our experiment. The use of water as a control in this kind of experiment where the effect of commercially formulated herbicides on beneficial insects are not uncommon (e.g. Hill et al., 2012, Biocontrol Science and Technology; Abraham et al., 2018, Entomologia Experimentalis et Applicata). For example, Hill et al. (2102) and Abraham et al. (2018) used water as control in their test of the effect of some commercially formulated herbicides on bees and some weevils (biocontrol agents of the invasive water hyacinth in South Africa). Therefore, we are confident that our usage of water as control is not aberration.

Smaller comments:

7. Collection of termites: When was the “Faculty of Life Sciences” established. This is relevant because the authors state no herbicides have been applied to the locality since then.

The Faculty of Life Sciences was established in June 2005.We have now inserted this date in the manuscript (see line 165 in new manuscript). 

8. Line 181. I assume the authors mean suspension rather than solution. I don’t know the exacts, but many commercial formulations of herbicides are suspensions and must be constantly mixed during application.

Many thanks. We have now replaced the word “solution” with “suspension”.

9. The discussion is too long. The section starting with line 327 can be dramatically shortened.

We have now deleted parts of the discussion and shortened it accordingly. (Please, see the deleted part in the revised manuscript with track changes). Because of the above action, reference number 65 has now been deleted.

Reviewer #2: 

I have inserted my comments through comments, please incorporate all the suggested changes. Once you incorporate the comments, then this article will be accepted. Discussion part is way longer, try to be concise and condense based results. Thanks

Many thanks for your comments. We have now deleted parts of the discussion section.

Responses to suggestions and comments in the annotated PDF 

Line 53: add Gill and Garg 2014 reference

Gill, H. K., and H. Garg. 2014. Pesticide: Environmental Impacts and Management Strategies, pp. 187-230. In: S. Solenski, and M. L. Larramenday (eds.). Pesticides- Toxic Effects. Intech. Rijeka, Croatia. 

We have now replaced reference number 2, Hussain et al. (2009) with “Gill and Garg (2014)”. This reference is a more recent and very germane 

Line 62: what this means? explain.

Pers. Obs. means “Personal observation”. We have now written this in full.

Line 65: write scientific names of crops. 

Attended to

Line 73: check spacing here

Attended to

Line 108: add scientific names

Because of the numerous species of dung beetles and dragon flies, we have elected to stick to the common name here and write out there scientific names when refereeing to one or a few species.

Line 135: refer to earlier comment, explain this. 

Pers. Obs. means “Personal observation”. We have now written this in full.

Line 171: add manufacturing company if have one.

Not known. We long bought this tiny brush in bunch and cannot remember the name of the manufacturing company

Line 201: applied manually or with any syringe

Applied with a syringe

Line 230: what kind of pencil?

We now inserted the “HB Pencil”

Lines 277 and 281:isn't "se" should be "SE" 

Done

Line 409: discussion is bit too much, keep all the references, but try condense .

Done

Line 525 - is missing

Nothing is missing. That is the format of writing the volume and page number of the journal

References cited in this rebuttal 

Abraham, J., Benhotons, G.S., Krampah, I., Tagba, J., Amissah, C. and Abraham, J.D. (2018). Commercially formulated glyphosate can kill non-target pollinator bees under laboratory conditions. Entomologia Experimentalis et Applicata 166:695-702.

Uyi, O.O., Zachariades, C. Heshula, L.U. and Hill, M.P. (2018). Developmental and reproductive performance of a specialist herbivore depend on seasonality of, and light conditions experienced by, the host plant. PLOS ONE 13(1): e0190700.

Lopez L Smith HA, Hoy MA and Bloomquist JR (2015). Acute toxicity and sublethal effects of fenpyroximate to amblyseius swirskii (acari: phytoseiidae). Jounal of Economic Entomology 108: 1047-1053.

Lukowski A, Giertych MJ, Zadworny M, Mucha J, Karolewski P (2015). Preferential feeding and occupation of sunlit leaves favors defense response and development in the flea beetle, Altica brevicollis coryletorum a pest of Corylus avellana. PLoS ONE. 2015a; 10(4): e0126072.

Hill, M.P., Coetzee, J.A and Ueckermann, C. (2012). Toxic effects of herbicides used for water hyacinth control on to insects released for its biological control in South Africa. Biocontrol Science and Technology 22(11):1321-1333.

---

## [Editor Report · Decision Letter 1]

6 Mar 2020

Exposure of the African mound building termite, Marcrotermes bellicosus workers to 2,4-D and atrazine based herbicides caused high mortality and impaired locomotor response

PONE-D-19-19112R1

Dear Dr. Uyi,

I would like to apologize for the long delay in processing your manuscript. Given the reservations of  Reviewer 1, I was hoping to have their re-review of your revised manuscript before rendering a decision. However, as we did not receive this re-review, I have carefully evaluated your revisions and find that the major reservations have been addressed, and the manuscript can therefore proceed.

We are pleased to inform you that your manuscript has been judged scientifically suitable for publication and will be formally accepted for publication once it complies with all outstanding technical requirements.

With kind regards,

Carla A Ng

Academic Editor

PLOS ONE
---

## [Editor Report · Acceptance letter]

10 Mar 2020

PONE-D-19-19112R1 

Exposure of the African mound building termite, *Macrotermes bellicosus* workers to commercially formulated 2,4-D and atrazine caused high mortality and impaired locomotor response 

Dear Dr. Uyi:

I am pleased to inform you that your manuscript has been deemed suitable for publication in PLOS ONE. Congratulations! Your manuscript is now with our production department. 

With kind regards,

on behalf of

Dr. Carla A Ng 

Academic Editor

PLOS ONE